# HDL in COVID-19 Patients: Evidence from an Italian Cross-Sectional Study

**DOI:** 10.3390/jcm10245955

**Published:** 2021-12-18

**Authors:** Bianca Papotti, Chiara Macchi, Chiara Favero, Simona Iodice, Maria Pia Adorni, Francesca Zimetti, Alberto Corsini, Stefano Aliberti, Francesco Blasi, Stefano Carugo, Valentina Bollati, Marco Vicenzi, Massimiliano Ruscica

**Affiliations:** 1Department of Food and Drug, University of Parma, 43124 Parma, Italy; bianca.papotti@unipr.it (B.P.); francesca.zimetti@unipr.it (F.Z.); 2Department of Pharmacological and Biomolecular Sciences, Università degli Studi di Milano, 20100 Milan, Italy; chiara.macchi@unimi.it (C.M.); alberto.corsini@unimi.it (A.C.); 3EPIGET Lab, Department of Clinical Sciences and Community Health, Università degli Studi di Milano, 20100 Milan, Italy; chiara.favero@unimi.it (C.F.); simona.iodice@unimi.it (S.I.); valentina.bollati@unimi.it (V.B.); 4Department of Medicine and Surgery, University of Parma, 43125 Parma, Italy; mariapia.adorni@unipr.it; 5IRCCS Multimedica, 20099 Sesto San Giovanni, Italy; 6Department of Biomedical Sciences, Humanitas University, 20100 Milan, Italy; stefano.aliberti@hunimed.eu; 7Fondazione IRCCS Ca’ Granda Ospedale Maggiore Policlinico, Internal Medicine Department, Respiratory Unit and Cystic Fibrosis Adult Center, 20100 Milan, Italy; francesco.blasi@unimi.it; 8Department of Pathophysiology and Transplantation, Università degli Studi di Milano, 20100 Milan, Italy; 9Fondazione IRCCS Ca’ Granda Ospedale Maggiore Policlinico, Cardiovascular Disease Unit, Internal Medicine Department, 20100 Milan, Italy; stefano.carugo@unimi.it; 10Dyspnea Lab, Department of Clinical Sciences and Community Health, Università degli Studi di Milano, 20100 Milan, Italy

**Keywords:** COVID-19, HDL, serum amyloid A, paraoxonase 1 activity, triglycerides

## Abstract

A number of studies have highlighted important alterations of the lipid profile in COVID-19 patients. Besides the well-known atheroprotective function, HDL displays anti-inflammatory, anti-oxidative, and anti-infectious properties. The aim of this retrospective study was to assess the HDL anti-inflammatory and antioxidant features, by evaluation of HDL-associated Serum amyloid A (SAA) enrichment and HDL-paraoxonase 1 (PON-1) activity, in a cohort of COVID-19 patients hospitalized at the Cardiorespiratory COVID-19 Unit of Fondazione IRCCS Ca’ Granda Ospedale Maggiore Policlinico of Milan. COVID-19 patients reached very low levels of HDL-c (mean ± SD: 27.1 ± 9.7 mg/dL) with a marked rise in TG (mean ± SD: 165.9 ± 62.5 mg/dL). Compared to matched-controls, SAA levels were significantly raised in COVID-19 patients at admission. There were no significant differences in the SAA amount between 83 alive and 22 dead patients for all-cause in-hospital mortality. Similar findings were reached in the case of PON-1 activity, with no differences between alive and dead patients for all-cause in-hospital mortality. In conclusion, although not related to the prediction of in-hospital mortality, reduction in HDL-c and the enrichment of SAA in HDL are a mirror of SARS-CoV-2 positivity even at the very early stages of the infection.

## 1. Introduction

Severe acute respiratory syndrome coronavirus 2 (SARS-CoV-2) infection has already left a permanent mark on human history. The virus, belonging to the family of Coronaviridae, primarily infects the respiratory tract, causing fever, sore throat, anosmia, and respiratory distress dyspnea. However, its tissue tropism is still not fully understood [1]. Besides causing respiratory disease, SARS-CoV-2 infection also associates to arterial hypertension [2] and cardiac injury, e.g., cases with and without classic coronary occlusion, arrhythmias, and heart failure [3]. In parallel, an overproduction of proinflammatory cytokines, e.g., tumor necrosis factor-alpha (TNF-*α*), interleukin 6 (IL-6), and IL-1*β*, has been described. Although so far there are no standard diagnostic criteria defining the onset of cytokine storm, in SARS-CoV-2-infected individuals, IL-6 surges during illness and declines during recovery [3,4]. Lipids and lipoproteins’ derangement, which play an important role in the pathogenesis and progression of atherosclerosis [5], have also been reported during bacterial, viral, and parasitic infections [6] specifically linked to TNF-*α* and IL-6 [7]. Consistently, since the beginning of the pandemic, a number of studies have highlighted important alterations of the lipid profile in COVID-19 patients. With this respect, as in the case of other systemic inflammatory conditions, a decrease in total cholesterol (TC), low-density lipoprotein (LDL)-cholesterol (LDL-c), and high-density lipoprotein (HDL)-cholesterol (HDL-c) levels has been reported [8]. Observational studies showed that low HDL-c levels could correlate with a higher risk of developing severe events in these patients [9,10]. An alteration in bioactive lipids has been also found during asymptomatic SARS-CoV-2 infection, possibly providing new tools for precise diagnosis [11].

In addition to the well-known atheroprotective function to promote reverse cholesterol transport, HDL displays anti-inflammatory, anti-oxidative, and anti-infectious properties, as well as anti-apoptotic and endotheliocytic protective effects, which may contribute to counteract viral and bacterial infections [12]. In patients with systemic inflammatory diseases, such as sepsis [13], autoimmune diseases [14], pneumonia [15], and other infections, alterations in the HDL functions have been reported, beyond the above-described decrement of HDL-c levels.

With respect to the HDL antioxidant activity, paraoxonase 1 (PON-1) is an esterase/lactonase enzyme almost exclusively associated with circulating HDL that protects HDL itself, LDL, macrophages, and endothelium from oxidation [15]. Low PON-1 activity has been associated with increased risk of major cardiovascular events, with many diseases characterized by a large inflammatory component (i.e., diabetes mellitus, rheumatoid arthritis, systemic lupus erythematosus, etc.) [16]. Beyond lower PON-1 activity, during acute and chronic inflammation HDL acquires a pro-inflammatory phenotype, which is characterized by serum amyloid A (SAA) enrichment. SAA is a highly sensitive acute phase reactant [17] that, during acute infection, can transiently increase > 1000-fold in the circulation [18].

While a growing number of studies [10,19] have focused on the evaluation of plasma HDL-c in COVID-19, few have evaluated HDL function. Thus, the aim of the present cross-sectional study was to assess the HDL anti-inflammatory and antioxidant features, by, specifically, SAA levels and PON-1 activity, in a cohort of COVID-19 patients hospitalized at the Cardiorespiratory COVID-19 Unit of Fondazione IRCCS Ca’ Granda Ospedale Maggiore Policlinico of Milan.

## 2. Materials and Methods

### 2.1. Patients

We enrolled 108 consecutive patients hospitalized at the Cardiorespiratory COVID- 19 Unit of Fondazione IRCCS Ca’ Granda Ospedale Maggiore Policlinico of Milan between February and June 2020. The present study was approved as a retrospective observational protocol by a local ethical committee (protocol n. 107162). In order to collect clinical data and biobanking, all patients signed an informed consent and approved their participation to NETwork Registry (advice n. 241_2020, protocol n. 101389) [20]. All patients signed an informed consent stored by the COVID-19 Network Working Group and the work was carried out in accordance with the Helsinki Declaration.

#### 2.1.1. Inclusion Criteria

Hospitalization within 24 h from the diagnosis of COVID-19 through a positive RT-PCR assay of nasopharyngeal swabs; respiratory failure was defined through arterial gas analysis and was expressed by partial pressure of oxygen to inspiratory fraction of oxygen ratio (PaO_2_/FiO_2_ < 400 mmHg). In all patients, body temperature, respiratory rate, systolic and diastolic blood pressure (BP, mmHg), and mean BP (calculated as 1/3 sBP + 2/3 dBP) were recorded.

#### 2.1.2. Patients’ Comorbidities Included in Data Set and Outcome

The following comorbidities were considered in the analysis: arterial hypertension, coronary artery disease, heart failure, peripheral vasculopathy, diabetes, chronic obstructive pulmonary disease, renal failure (defined as an estimated glomerular filtration rate < 60 mL/min/1.73 m^2^ calculated using the MDRD equation) and obesity (defined by a body mass index ≥ 30 kg/m^2^). Pre-existent cardiovascular active treatments including ACE inhibitors (ACE-i), angiotensin receptors antagonists, beta-blockers, statins, antiplatelet agents, and mineralocorticoid receptor antagonists (MRA) were recorded. Patients were treated according to the clinical protocol previously published [21]. All-cause in-hospital mortality was considered as a negative outcome.

### 2.2. Matched-Control Group

A group of subjects without SARS-CoV-2 infection was selected among individuals of the SPHERE study. Sixty-five subjects matched for age and BMI with COVID-19 patients were randomly selected [22]. The individual matching of patients with and without disease was not feasible; therefore, we ensured that the overall distribution of the confounding variables was similar. To reduce confounding, we frequently matched positive and negative subjects, balancing mean age and BMI in both groups. Gender matching was not possible due to differences in the frequency distribution in the two cohorts: The SPHERE study showed a higher prevalence of women, while the cohort of hospitalized COVID-19 patients showed a higher percentage of men. Participants were recruited between September 2010 and March 2015. Each participant signed an informed consent form, which had been approved by the ethics committee of the institution (approval number 1425), in accordance with the principles of the Helsinki Declaration.

### 2.3. HDL Isolation

The HDL fraction was isolated from COVID-19 and control sera by precipitating apoB-containing lipoproteins, as previously described [23]. Briefly, 40 parts of a polyethylene glycol (PEG) solution composed by 20% PEG 6000 (Sigma Aldrich, St. Louis, MO, USA) in 200 mM glycine buffer, pH 7.4, were added to 100 parts of serum. The solution was incubated for 20 min at room temperature and then spun for 30 min at 9500 g, at 4 °C. Finally, the supernatant containing the HDL fraction was collected and stored at −80 °C until use. This method allowed measuring the majority of serum SAA, namely, the HDL-associated SAA (approximately 95%) and the negligible amount of SAA not associated to HDL [17].

### 2.4. SAA Content in HDL Fraction

SAA was quantified in the isolated HDL serum fraction by colorimetric ELISA assay (Human Serum Amyloid A ELISA KIT—Sigma Aldrich), following manufacturer’s instructions, with a minimum detection range of 500 pg/mL. Human SAA concentration was expressed as ng/mL [13].

### 2.5. PON-1 Activity in HDL Fraction

HDL-bound PON-1 activity was measured in the isolated HDL serum fraction using the commercially available fluorometric PON-1 Activity Assay Kit (BioVision, Milpitas, CA, USA), according to the manufacturer’s guidelines. PON-1 activity was recorded though multiple kinetic measures for 60 min and expressed as (µU/mL), where 1 Unit of PON-1 activity corresponded to the amount of the enzyme that generated 1 µmol of fluorescent product per minutes at 37 °C and pH 8 [24].

### 2.6. Statistical Analysis

Data were evaluated by standard descriptive statistics. Continuous variables are expressed as mean ± SD, or as the median and interquartile range (Q1–Q3), as appropriate. Categorical variables are presented as absolute numbers and frequencies. Normality assumption was verified by graphical inspection. Baseline characteristics by SARS-CoV2 infection (yes vs. no) were reported. One-way ANOVA (analysis of variance) and one-way ANCOVA (analysis of covariance) analysis were used to test the relation between SAA and PON-1 levels and SARS-CoV2 infection status. The dependent variables were SAA or PON-1, and the independent variable was SARS-CoV-2 infection. To avoid confounding, all multivariable models were adjusted for gender and smoking habits. Models were also adjusted for BMI and age as a very slight variability between positives and negatives remained after frequency matching. The model testing the association of infection with SAA was also adjusted for antihypertensive drug use. The dependent variables were log-transformed to achieve normality of models’ residuals and results were reported as geometric means with a 95% confidence interval (CI). To examine the potential effect modification of age, we added the interaction term SARS-CoV2 infection * age to the one-way ANCOVA models. We evaluated whether the geometric mean of SAA concentrations by SARS-CoV2 infection status depended on age levels. Only in patients with SARS-CoV2 infection were one-way ANOVA and one-way ANCOVA analysis used to test the relation between SAA and PON-1 concentrations and survival status. All statistical analyses were performed with SAS software (version 9.4; SAS Institute Inc., Cary, NC, USA). A two-sided *p*-value of 0.05 was considered statistically significant.

## 3. Results

### 3.1. Characteristics of Patients and Matched-Controls

About 70% of COVID-19 patients were male and the median period from symptoms to diagnosis was of 10 (7–15) days. Mean age was 60 ± 3.1 years, with 24% of patients being overweight (mean BMI = 27.9 ± 4.3 Kg/m^2^) (Table 1). Hypertension represented the most common comorbidity (45.4%) with a total of 26 patients on renin-angiotensin-aldosterone system (RAAS) inhibitors (17 ACE-i and 9 ARBs), 16 were taking *β*-blockers, 13 were taking statins, 9 were taking acetylsalicylic acid, and none was taking mineral-receptor antagonists. Compared to the matched-control group, statistical differences were found in the lipid profile. SARS-CoV-2 infection led to a dramatic drop in the levels of TC (−33%), LDL-c (−37%), HDL-c (−55%), and non-HDL-c (−24.5%), with a concomitant rise in the levels of TG (+42%). In COVID-19 patients, we found that TG levels positively correlated with whole white blood cell count (*β* = 0.004, 95%CI 0.0008;0.007, *p* = 0.0124), lymphocytes’ count (*β* = 0.008, 95%CI 0.002;0.01, *p* = 0.009), and basophils (*β* = 0.6, 95%CI 0.12;1.08, *p* = 0.0148). Conversely, no associations were found between TG levels and monocytes, eosinophils, and neutrophils (Appendix A). White blood cell count is reported in Appendix A.

### 3.2. SARS-CoV-2 Infection Raises SAA Levels and PON-1 Activity

Analysis of the whole cohort (*n* = 173 individuals) showed a skewed distribution for both SAA content and PON-1 activity (Figure 1).

Relative to SAA, COVID-19 patients had significant geometric mean higher levels compared to matched controls, respectively, 57.8 (95% CI: 50.2;66.5) ng/mL and 32.7 (95% CI: 27.3;50.2) ng/mL. This difference was significant in both univariate analysis and upon correction for age, gender, BMI, and smoking (Table 2). SAA levels and PON-1 activity were lower in the control cohort compared to COVID-19 patients. This effect was maintained both in the univariate and adjusted models for age, gender, BMI, and smoking (Table 2).

To assess the possible modifier effect of age on the above reported association, we tested the possible interaction between age and SARS-CoV-2 infection. The difference of SAA mean levels between the COVID-19 group and the matched-controls was modified by age. Interestingly, the SAA adjusted geometric means decreased in controls but remain elevated in COVID-19 patients (Table 3).

Finally, in the whole cohort, SAA resulted to be negatively associated with BMI: Every rise in BMI of 1 Kg/m^2^ led to a decrement in SAA levels of 3.94% (Δ% = (exp(*β*) − 1) × 100 = −3.94, 95% CI: −6.31; −1.51, *p* = 0.0016). Relative to PON-1 activity, this was inversely associated with BMI: Every rise of 1 Kg/m^2^ in BMI, PON-1 activity was decreased by 2.69% (Δ% = (exp(*β*) − 1) × 100 = −2.69, 95% CI: −4.63; −0.71, *p* = 0.0080).

### 3.3. Survival Status, SAA Levels, and PON-1 Activity

In 108 patients with SARS-CoV2 infection, there were no significant differences in SAA amount and PON-1 activity between 83 alive and 22 dead patients for all-cause in-hospital mortality (Table 4).

## 4. Discussion

The main findings of this cross-sectional study are two-fold, namely, patients with SARS-CoV-2 infection have low levels of HDL-c (mean ± SD: 27.1 ± 9.7 mg/dL) with a marked rise in TG (mean ± SD: 165.9 ± 62.5 mg/dL), and second, HDL lipoproteins were enriched in SAA with a concomitant rise in PON-1 activity, with this last being surprising evidence.

Several observational studies have found that low levels of TC, LDL-c, or HDL-c levels are associated with an increased risk of developing infections and sepsis [25,26,27]. Relative to cholesterol metabolism, the inflammatory storm associated to infection decreases serum cholesterol as a result of a decrement in LDL-c. This is possibly due to an increase in the expression and activity of the LDL receptor and increased in LDL catabolism [28,29].

During infection, changes in the inflammatory milieu are characterized by a rise in TG, a phenomenon owing to different mechanisms: (1) an increase in the production of very-low-density lipoproteins or a decrement in their clearance, (2) an increase in the lipolysis of adipose tissue, (3) a rise in the de novo synthesis of hepatic fatty acid, and (4) the suppression of fatty acid oxidation [30]. In our cohort, TG plasma levels inversely and strongly correlated with HDL-c levels (*β* = −2.46, 95%CI −3.6; −1.3, *p* < 0.0001), as previously observed in HIV patients [31]. The combination of low HDL-c and high TG concentrations, measured before or during hospitalization, has been reported as a strong predictor of COVID-19 severity [32]. The liaison between blood TG and inflammation [33] may be due to different mechanisms, involving, among others, leukocytes’ activation [34]. This hypothesis is in line with the positive association we found between TG and whole white blood cell count. Data from UK Biobank confirmed this evidence to be peculiar in the case of an inflammatory milieu driven by infection and not specifically related to COVID-19 [35]. Indeed, TG levels did not associate either with COVID-19 infection [36] or with the risk of testing positive for SARS-CoV-2 infection [37]. Finally, it is worth mentioning that fatty acid can generally influence T cell activation, proliferation, and polarization [38,39].

During infection and inflammation, a marked decrease in serum levels of HDL-c and apolipoprotein (apo) A1 occurred [30]. Specifically, epidemiological studies showed that HDL-c levels reached very low levels, of 10 mg/dL in sepsis [40]. Concerning the liaison between HDL-c levels and infections, genetic studies provided evidence that among 97,166 individuals from the Copenhagen General Population Study, low HDL-c increased the risk of infection [41]. Similar conclusions were reached in the case of an analysis of 407,558 subjects of UK Biobank, providing causal inference for an inverse correlation among an increase of an HDL-c polygenic score and a reduction in the risk of hospitalization for infections and sepsis [42]. These observations, although not addressing potential mechanisms, provide strong evidence that HDL, quantified as HDL-c, is causally linked to host defense mechanisms against infections [43].

Conversely, in the case of SARS-CoV-2 infection, a bidirectional link between HDL-c and COVID-19 infection has been hypothesized [8]. Data from the UK biobank have shown that every 10 mg/dL increase in serum HDL-c or apo A1 levels corresponded to a 10% reduction in the risk of SARS-CoV-2 infection, after adjustment for age, sex, obesity, hypertension, type 2 diabetes, and coronary artery diseases [44]. Moreover, HDL-c seems to be associated with reduced odds of testing positive for SARS-CoV-2 [37]. Of note, very low levels of HDL-c have been found in the present study in which COVID-19 patients had a concentration of roughly 27 mg/dL compared to matched-controls (60.4 mg/dL). Although the sera of these latter were collected years before the COVID-19 outbreak, it is important to highlight that HDL-c and apo A1 levels do not change over time among middle-aged adults and are not strongly affected by environmental factors, as shown in a 35-year trajectory evaluation of lipids in the Framingham Heart Study [45].

Besides reducing HDL-c, inflammation drives marked changes in HDL-associated proteins, with some increasing and others decreasing. The most profound change is an enrichment in SAA and a decrease in apoA1 [46]. The SAA family of proteins is a major acute-phase reactant in mammals. During inflammation the plasma levels of SAA can transiently increase more than 1000 folds, counting the liver for the majority of circulating SAAs. SAA, which is not incorporated into HDL during HDL biogenesis [47], binds to HDL; with marked increases, it can displace apoA1 from HDL, thus becoming the major protein associated with HDL [48]. The enrichment of SAA in HDL impairs not only the HDL anti-inflammatory activity but promotes HDL degradation, possibly explaining the dramatic reduction in HDL-c levels [49]. A proteomic study comparing HDL proteins of critically ill COVID-19 patients showed both a rise in SAA family proteins and HDL that were unable to inhibit apoptosis caused by tumor necrosis factor *α* in endothelial cells [50]. Relative to the COVID-19 infection, SAA might be a useful indicator of disease severity with descending levels associated with a better prognosis compared to patients with an ascending trend [51,52]. In line with this background, although we found raised SAA levels in COVID-19 patients at admission compared to matched-controls, there were no significant differences in SAA amount between alive and dead patients for all-cause in-hospital mortality. This conclusion was confirmed in a proteome analysis of HDL, which showed that although SAA was increased by more than 50% in hospitalized COVID-19 patients, it was not a precise predictor of death [53]. Interestingly, we found that the difference of SAA between the COVID-19 group, and the matched-controls was modified by age; the delta between COVID-19 patients and controls was higher in those with an averaged mean age of 72 years compared to those with a mean age of 46 or 59. This was in agreement with the knowledge that age is an exponential risk factor of an adverse outcome in COVID-19 [54].

Another important aspect to take into consideration is the PON-1 activity, which is an HDL-associated enzyme that protects LDL from oxidation. The depletion of PON-1 decreases the antioxidant function of HDL while the addition of PON-1 increases the antioxidant function of HDL [55]. Surprisingly, although previous reports have shown that COVID-19 patients were characterized by a lower PON-1 activity and a rise in total PON-1 concentration [56], we found higher PON-1 activity in COVID-19 patients compared to controls. We had to consider that our patients were hospitalized within 24 h from the diagnosis and, thus, could not be considered in an advanced stage of the disease. Indeed, PON-1 was found to be less associated with HDL when evaluated in individuals hospitalized in ICU and treated with advanced respiratory support [50]. A further support to our hypothesis comes from the observation that the elevation in SAA levels, in our patients, was moderate (roughly 40%) instead of dramatically surging (up to 278%) as previously observed in severe COVID-19 patients [52]. This may hypothesize that the increase of PON-1 that we observed was temporary, mainly related to a first-line response against excess inflammation and oxidative stress. Among COVID-19 patients, PON-1 activity was inversely associated with BMI and significantly lower in those with a diagnosis of diabetes (geometric mean: 1299 µU/mL 95% CI 1044; 1615 vs. 1639 µU/mL 95% CI 1485; 1810) [57] or hypertension (geometric mean: 1419 µU/ML 95% CI 1238; 1626 vs. 1709 µU/ML 95% CI 1516; 1928) [58]. Similar to SAA, there were no significant differences in PON-1 activity between 83 alive and 22 dead patients for all-cause in-hospital mortality.

These data should be interpreted in a frame of limitations. First, this study was a retrospective and single-center design. Second, data were observational with no possibility to infer any causal mechanism. Third, the isolation of HDL fraction was made by precipitating apoB-containing lipoproteins and not by ultracentrifugation. However, this procedure has been utilized before to evaluate PON-1 activity and SAA levels during acute-phase reactions [13,59]. In addition, the precipitation of apoB-containing lipoproteins from serum preserved the activity of PON-1, as previously demonstrated [60], ruling out a possible interference of this method to isolate HDL on PON-1 measurements. Finally, we did not evaluate the levels of apoA1.

## 5. Conclusions

These findings show that, although not related to the prediction of all-cause in-hospital mortality, the reduction in HDL-c and the enrichment of SAA in HDL may be a mirror of SARS-CoV-2 positivity even at very early stages of the infection.

## Figures and Tables

**Figure 1 jcm-10-05955-f001:**
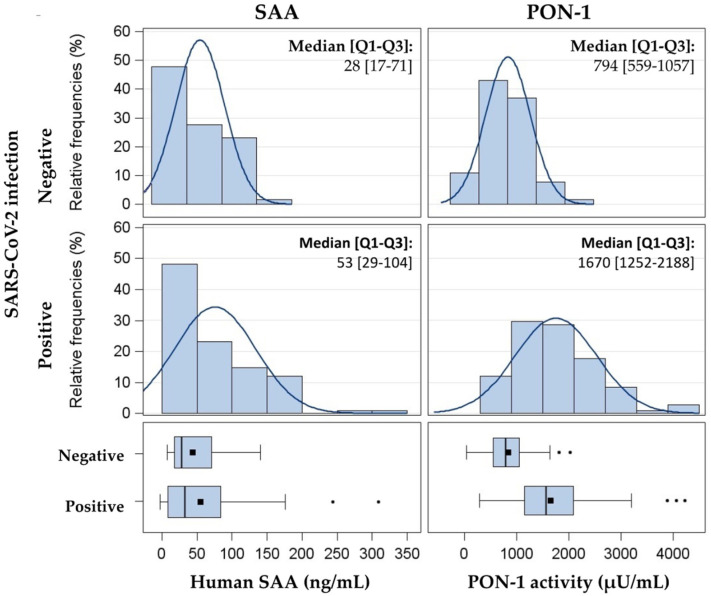
Distribution of SAA (ng/mL) and PON-1 activity (µU/mL) in 173 subjects. Data are presented as histograms and box-plots by SARS-CoV2 positivity. SAA, Serum amyloid A; PON-1, paraoxonase-1.

**Table 1 jcm-10-05955-t001:** Patients’ general characteristics, and clinical parameters at baseline.

	SARS-CoV2 Infection	
	Negative (*n* = 65)	Positive (*n* = 108)
Age, years	58 ± 12.9	60 ± 13.1
Gender		
Male	32 (49.2%)	75 (69.4%)
Female	33 (50.8%)	33 (30.6%)
BMI, kg/m^2^	28.2 ± 4.9	27.9 ± 4.3
Smoke		
Yes	5 (7.7%)	3 (2.8%)
No	60 (92.3%)	90 (83.3%)
Missing	-	15 (13.9%)
TC, mg/dL	207.3 ± 35.9	138.1 ± 46.5
LDL-c, mg/dL	123.8 ± 6.6	78.0 ± 24.6
HDL-c, mg/dL	60.4 ± 14.3	27.1 ± 9.7
non-HDL-c, mg/dL	147.0 ± 36.8	110.9 ± 44.5
Triglyceride, mg/dL	116.7 ± 72.5	165.9 ± 62.5
Antihypertensive medications		
Yes	11 (16.9%)	16 (14.8%)
No	54 (83.1%)	92 (85.2%)

BMI, body mass index; TC, total cholesterol; LDL-c, low-density lipoprotein cholesterol; HDL-c, high-density lipoprotein cholesterol.

**Table 2 jcm-10-05955-t002:** Differences in SAA levels and in PON-1 activity by SARS-CoV2 positivity.

Outcome	SARS-CoV2 Infection	Univariate Model		Adjusted Model	
		Mean (95% CI)	*p*-Value	Mean (95% CI)	*p*-Value
SAA (ng/mL)	Negative	32.7 (27.3; 50.2)	<0.0001	35.9 (27.3; 47.2)	<0.0001
	Positive	57.8 (50.2; 66.5)		60.5 (45.4; 80.5)	
PON-1 activity (µU/mL)	Negative	702.5 (612.8; 805.3)	<0.0001	618.3 (502.1; 761.4)	<0.0001
	Positive	1579.3 (1420.5; 1755.8)		1394.0 (1123.8; 1729.2)	

Data are expressed as geometric means and 95% confidence interval. Data are adjusted for age, gender, BMI, smoking, and antihypertensive medications (the latter only for SAA). SAA, serum amyloid A; PON-1, paraoxonase-1.

**Table 3 jcm-10-05955-t003:** Interaction effect among age and SARS-CoV2 infection on SAA levels.

Outcome	Age	SARS-CoV2 Infection	Mean (95% CI)	*p*-Value	*p*-Value for Interaction
SAA (ng/mL)	mean − SD: 46 years	Negative	47.1 (34.3; 64.7)	0.1780	0.0087
		Positive	58.3 (42.4; 80.4)		
	mean: 59 years	Negative	34.4 (26.2; 45.0)	<0.0001	
		Positive	59.3 (44.8; 78.5)		
	mean + SD: 72 years	Negative	25.1 (17.7; 35.5)	<0.0001	
		Positive	60.3 (44.3; 82.3)		

Association between SAA and case control group was evaluated at three selected levels of age (mean-standard deviation (SD), mean, and mean + SD value). Data are expressed as adjusted geometric means and 95% confidence interval. Data are adjusted for gender, BMI, smoking, and antihypertensive medications. SAA, Serum amyloid A.

**Table 4 jcm-10-05955-t004:** Association between survival status and SAA or PON-1 activity in patients with SARS-CoV2 infection.

Outcome	Survival Status	Univariate Model		Adjusted Model	
		Mean (95% CI)	*p*-Value	Mean (95% CI)	*p*-Value
SAA (ng/mL)	Death	60.1 (44.2; 81.7)	0.8616	64.3 (37.8; 109.3)	0.9530
	Alive	58.3 (49.8; 68.3)		63.4 (40.2; 99.9)	
PON-1 activity (µU/mL)	Death	1512.1 (1237.5; 1847.6)	0.6523	1687.4 (1340.2; 2124.4)	0.5480
	Alive	1592.5 (1436.4; 1765.5)		1558.0 (1398.5; 1735.6)	

Data are expressed as geometric means and 95% CI. Data are adjusted for age, gender, BMI, smoking, and antihypertensive medications (the latter only for SAA). SAA, Serum amyloid A; PON-1, paraoxonase-1.

## Data Availability

The references included in this study are available from the online database or on request from the corresponding author.

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
