# Peer review of "HDL in COVID-19 Patients: Evidence from an Italian Cross-Sectional Study"

_jcm, 2021, doi:10.3390/jcm10245955_

Round 1
Reviewer 1 Report
In this retrospective, observational, monocentric study, the authors investigated lipid parameters as well as SAA and PON-1 levels in COVID-19 patients admitted to their clinic over a period of February to June 2020. The authors show significant changes with the reduction in HDL-c and the increase in the TG levels as well as elevated SAA and PON-1 in COVID-19 patients compared to the control group. Levels of SAA and PON-1 were, however, not predictive for the disease outcome. The authors also acknowledge possible limitations of their study, including study design, technical and methodological aspects as well as limitations in data availability.
There are, however, a few remaining questions regarding this study.
Major concerns:
- The reviewer finds possible caveats in using the proposed control group for several reasons:
- As authors correctly notice and mention in the discussion, the control group samples were collected 5-10 years before the COVID-19 outbreak. Although the authors refer to a study demonstrating the stability of ApoA1 and HDL over a prolonged period of time, there is little evidence that the same applies for SAA or activity of PON-1. This is particularly problematic regarding the PON-1 activity since it may be the reason why the values are lower in the control group compared to the diseased individuals, whereas the opposite was reported in the past (reference 50)
- The inclusion criteria for the control group took into account age and the BMI (which is understandable, since the original SPHERE study recruited overweight and obese participants). However, gender (and smoking, as found in the reference 41) may have a major influence on the plasma lipid levels, and gender imbalance is quite prominent in the COVID-19 compared to the control group. Since the gender-specific frequency of SARS-CoV2 infection is not the topic of this manuscript, gender matching should also be performed. Moreover, authors report use of some medication including lipid-lowering and/or antihypertensive drugs and their contribution to the results should also be taken into account, since they can significantly influence plasma lipid levels and functional characteristics of HDL.
- Would it not be better to have a COVID-19 negative group as a control, which is matched with age, gender, BMI, comorbidities, and medication? Although it may be much smaller, it could be introduced in addition to the existing SPHERE control group.
- The figure is not clearly presented. Do histograms represent the distribution in controls and the curve in COVID-19 patients? Why not representing both groups with a curve/histogram in a different color and use the same color pattern in the box-plot below?
- The discussion part mentions a correlation between serum TG and lymphocytes. This data should be presented in the results section, expand to all investigated blood cells, and appropriately discussed according to the current literature on COVID-19.
- This article shares some similarities with another one, which the reviewer was not able to find among the references:
https://doi.org/10.1038/s41598-021-81638-1
The authors should not for get to cite the mentioned study, particularly since it analyzed HDL on days 1, 3, and 7 upon hospitalization. Moreover, the mentioned study finds reduced PON1 in HDL of COVID-19 patients and suggests that neutrophil activation in COVID-19 patients causes the release of elastase and concomitant proteolytic clearance of HDL-associated PON-1. These data are in sharp contrast to the results of the current study and the authors should discuss the discrepancies of the two studies in detail.
Minor concerns:
- The results section should be presented in a more coherent way. The text goes from Table 2 to Table 3 back to Table 2. It should have a logical flow without going back and forth.
- The authors also mention that the isolation of HDL was performed with ApoB-lipoprotein precipitation and not by ultracentrifugation. This would mean that the supernatant upon removal of ApoB lipoproteins contains all non-ApoB serum components. As the SAA is mainly (approx. 95%) but not entirely associated with HDL, a small amount of SAA not associated with HDL would be measured in the sample prepared in this way. This should be suggested in the text.
- The data should be presented as control first and COVID-19 patients as the second, in the figure as well as in the tables
- SARS-CoV-2 is a pathogen and COVID-19 is the disease caused by the SARS-CoV-2. Therefore, it cannot be “SARS-CoV-2 patients” (which appears in the abstract and several times in the text), but SARS-CoV-2 infection and COVID-19 patients.
- There is an obvious shift in style and quality of the data interpretation between rows 262 and 301. This section needs to be rewritten.
- Line 268: “In our cohort, 266 TG plasma levels inversely and strongly correlated with HDL-c levels (= -2.46, 95%CI -267 3.6;-1.3, p <0.0001), as previously observed [30].” Please specify in which disease, since the reference does not handle COVID-19.
Author Response
Thnak you for your constructive comments. Please find enclosed the pdf file with alla the answers.

Reviewer 2 Report
Introduction
The authors might want to consider describing what the outcome is in the last paragraph of the introduction. They successfully address what the exposure is (HDL measures), but not the outcome (severe covid? Death?). It is important for the reader to know what the outcome is.
Methods
- Maybe it would be beneficial to divide the patient’s section. This section includes data on outcome and exposure variables. Authors can consider having a section of outcome and exposure to define these variables. Further, the authors can consider explaining inclusion and exclusion criteria in this same section.
- If this is a case-cohort study, authors should address more on how they selected the controls for the study in the matching section. Why did they only matched on age and BMI and not age and sex and BMI? Sex can be a potential confounder here and authors should consider it. Further, did the authors use a frequency matching or individual matching. They should describe this as well.
- Authors should consider depicting DAGs to show the matching process and the analysis process. How they matched on covariates and what analysis they will perform based on this matching. Will they incur in collider bias if adjusting on a variable that is not necessary to match? This will benefit the readers in understanding how the authors addressed selection bias for example.
Results
- P-values are not necessary in table 1. Authors should consider omitting them (essentially because they matched the groups and so knowing potential differences is not necessary due to the nature of the study). This misleads the readers.
Author Response
Reviewer#2
The authors might want to consider describing what the outcome is in the last paragraph of the introduction. They successfully address what the exposure is (HDL measures), but not the outcome (severe covid? Death?). It is important for the reader to know what the outcome is.
Methods
- Maybe it would be beneficial to divide the patient’s section. This section includes data on outcome and exposure variables. Authors can consider having a section of outcome and exposure to define these variables. Further, the authors can consider explaining inclusion and exclusion criteria in this same section.
Thank you for your suggestion. We have tried to make this section clearer.
- If this is a case-cohort study, authors should address more on how they selected the controls for the study in the matching section. Why did they only matched on age and BMI and not age and sex and BMI? Sex can be a potential confounder here and authors should consider it. Further, did the authors use a frequency matching or individual matching. They should describe this as well.
We thank the reviewer for raising this comment which allowed us to better specify the study design and the matching variables. We performed a cross-sectional study, comparing HDL levels in subjects with and without SARS-CoV-2 infection. Positive subjects were selected from a cohort of hospitalized patients with a diagnosis of COVID-19 through nasopharyngeal swabs, while negatives were selected from the cohort of SHERE patients. The individual matching of patients with and without disease was not feasible, so we ensure that the overall distribution of the confounding variables was as similar as possible. To reduce confounding, we frequently matched positive and negative subjects with the attempt to balance mean age and BMI in both groups. Although highly desirable, gender matching was not possible due to differences in the frequency distribution in the two cohorts: the SPHERE study showed a higher prevalence of women, while the COVID-19 cohort was made up of a high percentage of men. To avoid confounding, we adjusted all multivariable models for gender, together with the other covariates.
- Authors should consider depicting DAGs to show the matching process and the analysis process. How they matched on covariates and what analysis they will perform based on this matching. Will they incur in collider bias if adjusting on a variable that is not necessary to match? This will benefit the readers in understanding how the authors addressed selection bias for example.
As described in the previous remark, we performed a cross-sectional study to evaluate the association between SARS-CoV-2 infection and SAA levels or PON-1 activity. This type of study has a limited ability to draw conclusions about causality because the presence of risk factors and outcomes are measured simultaneously. Causal diagrams depict a theoretical causal structure for exposure to the outcome, therefore we believe that it is not conceptually consistent with the results.
In the present study, the measures of effect are the marginal means of SAA levels or PON-1 activity in patients with and without SARS-CoV-2 infection, obtained from ANCOVA models. The dependent variables are SAA levels or PON-1 activity, and the independent variable is SARS-CoV-2 positivity. In this context, colliding bias is not inherent with design and the outcomes. To avoid other sources of possible bias, such as confounding, we adjusted all multivariable models for gender and smoking habits. We also adjusted models for BMI and age as a very slight variability between positive and negatives remained after frequency matching. Model testing the association with SAA levels was also adjusted for antihypertensive drug use. To examine the potential effect modification of age, we also performed a model adding the interaction term SARS-CoV2 infection*age.
We expanded the method section to help the understanding of the study design. We also changed the manuscript title by removing “cohort”. It now reads as follows “HDL IN COVID-19 PATIENTS – EVIDENCE FROM AN ITALIAN CROSS-SECTIONAL STUDY”
Results
- P-values are not necessary in table 1. Authors should consider omitting them (essentially because they matched the groups and so knowing potential differences is not necessary due to the nature of the study). This misleads the readers.
We thank the reviewer for her/his suggestion. P values have been removed.

Round 2
Reviewer 1 Report
The authors amended all concerns and thoroughly revised the critical points.
Minor corrections:
- Use of “Oxford comma” should be consistent throughout the text, minor spell check required
- Rows 295-297: the sentence is not clear and requires rephrasing
Author Response
Use of “Oxford comma” should be consistent throughout the text, minor spell check required
Thank you. The manuscript has been revised accordingly.
Rows 295-297: the sentence is not clear and requires rephrasing
We thank the reviewer for point this out. The sentence has been re-wrote as follows "Of note, very low levels of HDL-c have been found in the present study in which COVID-19 patients had a concentration of roughly 27 mg/dL compared to matched-controls (60.4 mg/dL)."
Reviewer 2 Report
Thank you for addressing the recommendations provided. No further comments.
Author Response
Thank you for addressing the recommendations provided. No further comments.
We thank the reviewer for her/his thoughtful comment.